# Accuracy of ankle-brachial index in screening for peripheral arterial disease in people with diabetes

**Monique Magnavita Borba da Fonseca Cerqueira**[1], **Neiva Sueli Santana Gonçalves Bastos**[2], **Dandara Almeida Reis da Silva**[1], **Dario Gregori**[3]*, **Lucélia Batista Neves Cunha Magalhães**[4], **Magno Merces Weyll Pimentel**[5]

**1** Department of Life Sciences, State University of Bahia, Salvador, BA, Brazil, **2** University Center Unidompedro, Salvador, BA, Brazil, **3** School of Medicine, University of Padova, Padova, Italy, **4** Zarns School of Medicine, Eunápolis, BA, Brazil, **5** School of Medicine, Federal University of Bahia, Salvador, BA, Brazil

* dario.gregori@unipd.it

**Data Availability Statement:** All relevant data are within the paper and its Supporting Information files.

## Abstract

Although the ankle-brachial index (ABI) presents overall satisfactory accuracy, its sensitivity in the context of screening strategies does not ensure the detection of all individuals with peripheral arterial disease (PAD), especially in clinical situations where there is calcification of the arterial media layer. This study evaluated the accuracy of ABI in screening PAD among individuals with diabetes mellitus (DM) in a community setting. An observational study included only individuals with DM. ABI measurement was performed, and the lower limb duplex ultrasound (DU) was used as the reference standard for PAD diagnosis. Sensitivity, specificity, positive and negative predictive values (PPV and NPV), and positive and negative likelihood ratios (LR+ and LR-) of ABI were assessed. The analysis included 194 limbs from 99 participants, with a PAD prevalence identified by DU of 15.98%. ABI demonstrated an accuracy of 87.63%, with a sensitivity of 35.48%, specificity of 97.55%, PPV of 73.33%, NPV of 89.83%, LR+ of 14.46, and LR- of 0.66. ABI showed high specificity but limited sensitivity in detecting PAD among individuals with DM in a community setting. An LR- of 0.66 suggests that a normal ABI result reduces but does not eliminate the possibility of PAD, highlighting the importance of complementary diagnostic approaches to enhance accuracy in identifying PAD in high-risk patients, such as those with DM. Incorporating additional diagnostic methods may be necessary to improve the effectiveness of PAD screening in this group.

## Introduction

The ankle-brachial index (ABI) is recognized as an essential diagnostic tool in the initial assessment of peripheral arterial disease (PAD), as well as a marker for generalized atherosclerosis and cardiovascular risk [1]. ABI measurement is recommended in patients presenting with symptoms suggestive of PAD or diagnosed with atherosclerotic disease elsewhere [2]. For

**Funding:** The Research Program for the Unified Health System PPSUS/BA 2020 7th edition number 4411/2020 funded the study. The funding organization had no role in the design, collection, analysis, or interpretation of the data or in the decision to submit the manuscript for publication.

**Competing interests:** The authors have declared that no competing interests exist.

asymptomatic individuals, guidelines recommend performing ABI from the age of 65 for the general population, from 55 for those classified as at high cardiovascular risk, and from 50 for those with a family history of PAD [2].

ABI values of 0.9 or lower suggest the presence of PAD; values between 0.9 and 1.4 are interpreted as standard, while those above 1.4 indicate arterial stiffness [2]. Arterial medial layer calcification, a condition prevalent among individuals of advanced age and those with diabetes mellitus (DM), chronic kidney disease, or hypertension, can impede the compression of foot arteries, leading to falsely elevated ABI values and underestimating the prevalence of PAD [3, 4].

DM is one of the most significant risk factors for the development of PAD. This condition, characterized by atherosclerotic formation in the arteries of the lower extremities, is a macro-vascular complication, indicating a risk for events such as ulceration and amputation. More-over, it is associated with more than double the rates of coronary events, cardiovascular mortality, and total mortality over 10 years [2]. In people with diabetes, the combination of PAD and peripheral neuropathy (PN) often results in subclinical manifestations of limb ische-mia. Therefore, the clinical management of these patients requires early and optimized diag-nostic and therapeutic interventions [5].

Although the index presents overall satisfactory accuracy, its sensitivity in the context of screening strategies fails to ensure the detection in a high number of individuals with PAD [6]. Remarkably, the test's sensitivity is limited in the DM population, rendering it inadequate as the sole screening tool for the disease [7]. Furthermore, the accuracy of an instrument depends not only on its sensitivity and specificity but also on the prevalence of the disease in the studied population [8]. When considering the findings of clinical studies for decisions in public health or clinical practice, the origin of participants is essential, as excessive selection may not repre-sent the general population, compromising the applicability of the results. This issue is espe-cially relevant in studies conducted in specialized centers, where disease prevalence and severity differ from the community population, impacting test performance improvement [8, 9].

The current evidence regarding the role of ABI in diagnosing PAD is based on a limited number of studies with moderate to high risks of bias [10]. Many studies on ABI accuracy included only symptomatic or asymptomatic patients already under follow-up in specialized centers, generally with more complicated diabetes, and did not consider the diversity of dia-betic populations in community settings [10]. Additionally, there were gaps in demographic information, frequently omitting data on peripheral neuropathy or active ulcerations [7].

Understanding the relevance of an appropriate diagnosis and management of atheroscle-rotic disease and considering the intrinsic limitations of ABI in screening for PAD in people with diabetes, especially in population settings, the present study aims to assess the accuracy of ABI in diagnosing PAD among individuals with DM in a community context. A careful analy-sis of the effectiveness of this index in this specific group can ground the discussion on the need to integrate complementary approaches, aiming for a more accurate diagnosis of arterial disease in subgroups with a high-risk profile.

## Material and methods

An observational study in a community setting was conducted per the Standards of Reporting of Diagnostic Accuracy Studies (STARD) [11].

Participants were recruited from a defined territory in Salvador, Bahia, Brazil, through research dissemination, with the support of the residents' association, between October 2021 and May 2023. Those aged 18 years or older with a diagnosis of DM, according to the

American Diabetes Association criteria [12] were included. Exclusion criteria included living outside the recruitment territory and having a history of previous lower limb revascularization. Those who met the inclusion criteria and signed the Informed Consent Form (ICF) were voluntarily and consecutively incorporated into the research, proceeding to the subsequent investigation phases in the Research Laboratory. Sample size calculation determined the need to examine a total of 262 limbs, considering the following parameters: a confidence level of 95%, ABI sensitivity of 65%, a margin of error of 10%, and a PAD prevalence of 30% among the 2450 locally registered individuals with diabetes [7, 13].

Clinical information influencing overall cardiovascular morbidity and lower limb conditions, as well as signs and symptoms that allowed PAD classification, were considered. Demographic and clinical data were collected through interviews and clinical examination using a semi-structured questionnaire developed by the study's authors. Self-reported comorbidities included systemic arterial hypertension (SAH), smoking, coronary artery disease (CAD), stroke, and stage 5 chronic kidney disease (5 CKD). Existing diabetic foot ulcers (DFUs) were documented, detailing their extent, depth, and presence of infection. Identified amputations were categorized by level.

Participants were classified according to symptoms of lower limb ischemia based on the Edinburgh Claudication Score [14]. Additionally, they were stratified by the Wound, Ischemia, and foot Infection (WIfI) classification, proposed by the Society for Vascular Surgery, defining the presence of ulcer, ischemia, and infection [15]. Protective plantar sensitivity was assessed using a 10g Semmes-Weinstein monofilament SORRI-BAURU®, applied at three predefined points on the feet. Sensitivity was considered present when the individual correctly identified the application in two out of three attempts and absent if they failed two out of three [16].

## Diagnostic tests

The index and reference tests were conducted by a single, trained, and certified specialist, with intervals of one to two weeks between them. Due to the impossibility of blinding the examiner or including another investigator, this strategy was adopted to minimize biases and ensure the consistency of results.

For the ABI measurement, participants remained supine for five to ten minutes before the examination. They were also asked to avoid consumption of alcohol, smoking, exercise, and caffeine one hour prior to the test. A 10 MHz Portable Vascular Doppler device, model DV 610, MEDMEGA®, and an aneroid sphygmomanometer, PREMIUM®, were used to measure the systolic blood pressures in the arms and calves bilaterally.

The index calculation was based on the ratio of the highest systolic pressure values found in the ankles to that in the arm. PAD was defined by an ABI <0.90, while an ABI >1.40 indicated arterial stiffness. Values between these limits were interpreted as normal [2].

The lower limb arterial duplex ultrasound (DU) was adopted as the standard method for PAD assessment, as it is a non-invasive, accessible, and accurate examination for identifying the disease [17]. DU allows for the detection and localization of vascular lesions, quantifying their extent and severity based on criteria of velocity and waveform morphology [18]. The examinations were performed using a linear transducer (5.0 to 12.0 MHz) on a Xario XG Toshiba® ultrasound device, with participants in a horizontal decubitus position, in a controlled temperature environment, resting for five to ten minutes before the examination. Similarly to the ABI measurement, activities and substances capable of influencing measurements were recommended to be avoided before data collection.

Significant PAD was defined, according to the Society for Vascular Medicine and the Society for Vascular Ultrasound criteria, by an increase in peak systolic velocity (PSV) greater than

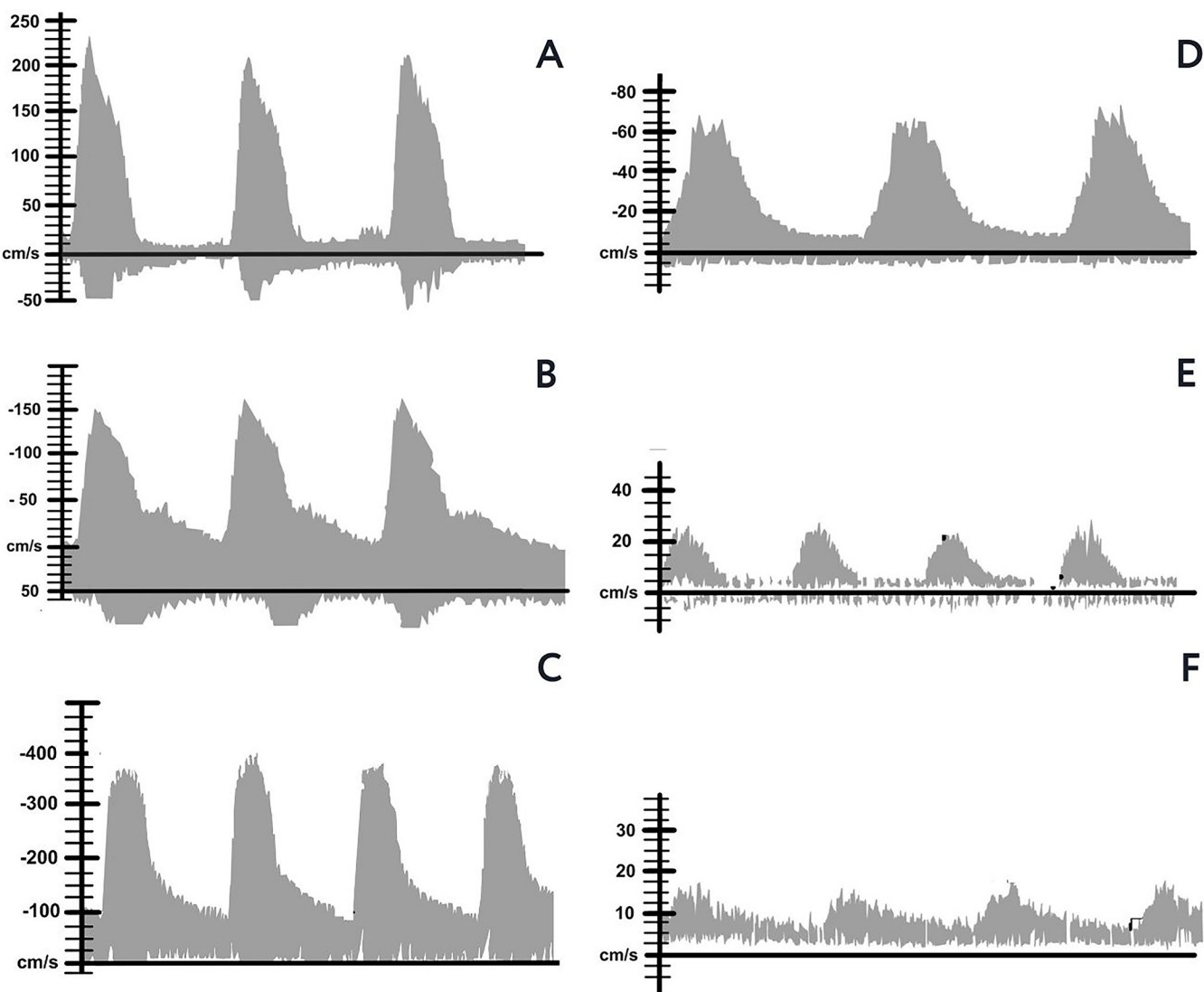

**Fig 1. Wave patterns in PAD. A** and **B**: stenosis > 50%. **C**: stenosis > 75%. **D**: continuous pandiastolic flow, delayed systolic rise, spectral broadening, **E** and **F**: monophasic flow [18].

100% compared to the regular proximal segment for stenoses above 50% (Fig 1A and 1B), and more than four times for stenoses above 75% (Fig 1C). Additionally, wave morphology compatible with a monophasic pattern, with continuous pandiastolic flow, delayed systolic upstroke, and spectral broadening (Fig 1D–1F), was also used as diagnostic criteria. The absence of flow indicated artery occlusion [18].

## Statistical analysis

A descriptive analysis characterized the sample in terms of sociodemographic and clinical aspects. Categorical variables were reported as frequencies and percentages, while continuous variables with a normal distribution were presented as means and standard deviations. The Kolmogorov-Smirnov test assessed normality. Variables with a non-normal distribution were expressed in the median and interquartile range. The software Statistical Package for Social

Sciences (SPSS® Inc., Chicago, IL, USA), version 17.0 for Windows®, was used to develop the database and analysis.

The ABIs found were compared with the reference method, the DU. In the accuracy analysis, ABI values above 1.4 were not included, as measures above this limit render the index an inappropriate diagnostic tool for identifying PAD in calcified arteries [19].

For the analysis of accuracy measures, sensitivity, specificity, positive predictive value (PPV), negative predictive value (NPV), positive likelihood ratio (LR+), and negative likelihood ratio (LR-) were calculated. Each limb was analyzed as an independent observation. All values were calculated using the exact Binomial approach [20].

A Bayesian regression model was implemented to explore the diagnostic performances of the ABIs on specific subgroups of the population. The model considered age, sex, race, CAD and SAH as subgroups of interest. Other variables of potential clinical interest were excluded because of insufficient numbers. The model was fitted using the brm function from the brms package in R, with the response modeled using a Bernoulli family [21, 22].

Normal priors with a mean of 0 and a standard deviation of 5 were assigned to the regression coefficients, while the intercept was given a Cauchy prior with a location of 0 and a scale of 2. These choices for priors are well-supported in the literature for their regularizing properties and ability to improve convergence in Bayesian models [23]. The MCMC sampling procedure involved 10,000 iterations per chain, with the first 2,000 iterations designated for warm-up across four chains. The posterior probability distribution of obtaining a given level of specificity and sensitivity was computed. An acceptable threshold for specificity of 80% was chosen to compute the probability of staying above it.

This research followed the ethical principles of the Declaration of Helsinki and was approved by the Ethics Committee under Ethics Appraisal Submission Certificate number 38514920.7.0000.0057 and decision number 4.327.233 on October 8, 2020. The Research Program for the Unified Health System PPSUS/BA 2020 7th edition number 4411/2020 funded the study.

## Results

A flowchart of the study design was developed, detailing the inclusion of participants at each phase (Fig 2). The average age of the sample was 58.5 years (±11.3), with a predominance of the female sex (76.2%). About three-quarters of the participants had been diagnosed with DM for less than ten years (75.2%). SAH emerged as the most common comorbidity (70.5%), while CAD and CVD were reported by 20.0% and 8.6% of the participants, respectively. Table 1 presents the demographic and clinical characteristics of the studied population.

Regarding lifestyle habits, 6.7% of participants were current smokers, and 22.8% had been smokers in the past. Regular consumption of alcoholic beverages was reported by 38.1% of the participants, with 30% of these indicating abuse or dependence. Treatment adherence was documented in only 15.2% of the studied population.

Intermittent claudication was observed in 7.6% of individuals. Two cases of DFU were recorded, corresponding to 1.9% of participants. Only 3.8% of participants had a history of prior amputation, with digital amputations predominating (75%). Regarding the WIfI system classification, the entirety of the sample was categorized as very low (93.9%) or low risk (6.1%). Protective plantar sensitivity was present in 72.4% of the studied population.

Four of the 105 individuals who agreed to participate in the study did not complete the index test due to withdrawal, and three were removed due to duplicate records. Of the 99 participants effectively subjected to the test, the ABI could not be measured in 4 limbs due to prior amputations (two limbs) and injuries at the measurement site (two limbs), resulting in

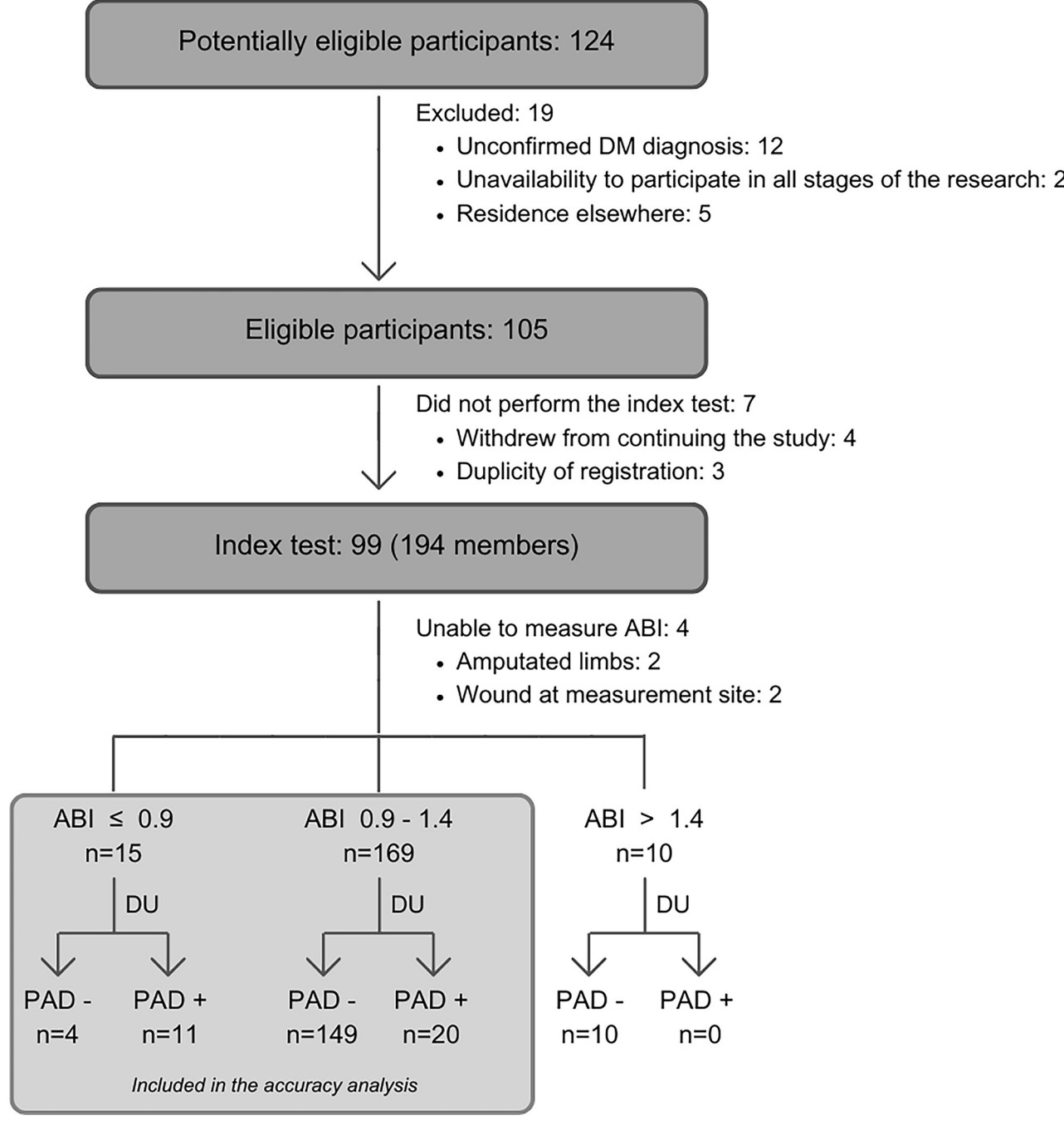

**Fig 2. Study flowchart.**

the analysis of 194 limbs. Among these, 87.1% exhibited ABI within the normal range (0.9–1.4), 7.7% indicated the presence of PAD, and 5.2% suggested arterial stiffness, including two cases of incompressible arteries. The mean ABI was 1.17 (± 0.2).

The DU examination on 194 limbs detected PAD in 15.98% of cases. The data analysis revealed that the test had 11 true positives, 4 false positives, 159 true negatives, and 20 false negatives. Table 2 presents the accuracy of ABI in identifying PAD among individuals with

**Table 1. Population characteristics.**

| Variables | N = 105 |
|---|---|
| Age mean ± SD[a] | 58,5 ± 11,3 |
| | n (%) |
| Woman/Men | 80 (76,2) / 25 (23,8) |
| Brown/ Black | 43 (41,0) / 58 (55,2) |
| Education | |
| Level 1 (illiterate/incomplete elementary school) | 43 (41,0) |
| Level 2 (complete elementary school/incomplete high school) | 22 (21,0) |
| Level 3 (complete high school/higher education) | 40 (38,1) |
| Family Income | |
| Up to 1 minimum wage[b] | 64 (61,0) |
| Between 1 and 2 minimum wages | 24 (22,9) |
| Above 2 minimum wages | 17 (16,2) |
| Time since DM diagnosis | |
| < 5 years | 48 (45,7) |
| 5 to 10 years | 31 (29,5) |
| 11 or more | 26 (24,8) |
| SAH | 74 (70,5) |
| Peripheral Systolic Blood Pressure | 144,0 ±21,2 mmHg |
| Peripheral Diastolic Blood Pressure | 83,4 ±10,7 mmHg |
| CAD | 21 (20,0) |
| CVD | 9 (8,6) |
| 5 CKD | 2 (1,9) |
| Current or former smoking | |
| No | 74 (70,5) |
| Ex-smoker | 24 (22,9) |
| Yes | 7 (6,7) |
| Alcohol consumption | 40 (38,1) |
| Alcohol abuse or dependence (CAGE) | 12/40 (30,0) |
| Non-therapeutic adherence (Morisky Green) | 89 (84,8) |

[a]SD = stantard deviation

[b]Brazilian national minimum wage corresponds to $200 in the United States of America.

**Table 2. ABI's accuracy.**

| Statistic | Value | 95% CI |
|---|---|---|
| Sensivity | 35.48% | 19.23% to 54.63% |
| Specificity | 97.55% | 93.84% to 99.33% |
| Positive Predictive Value* | 73.33% | 49.01% to 92.21% |
| Negative Predictive Value* | 88.83% | 83.27 to 93.04% |
| Positive Likehood Ratio | 14.46 | 4.92 to 42.49 |
| Negative Likehood Ratio | 0.66 | 0.51 to 0.86 |
| Accuracy* | 87.63% | 82.15 to 91.91% |
| Disease prevalence | 15.98% | |

* These values are dependent on disease prevalence.

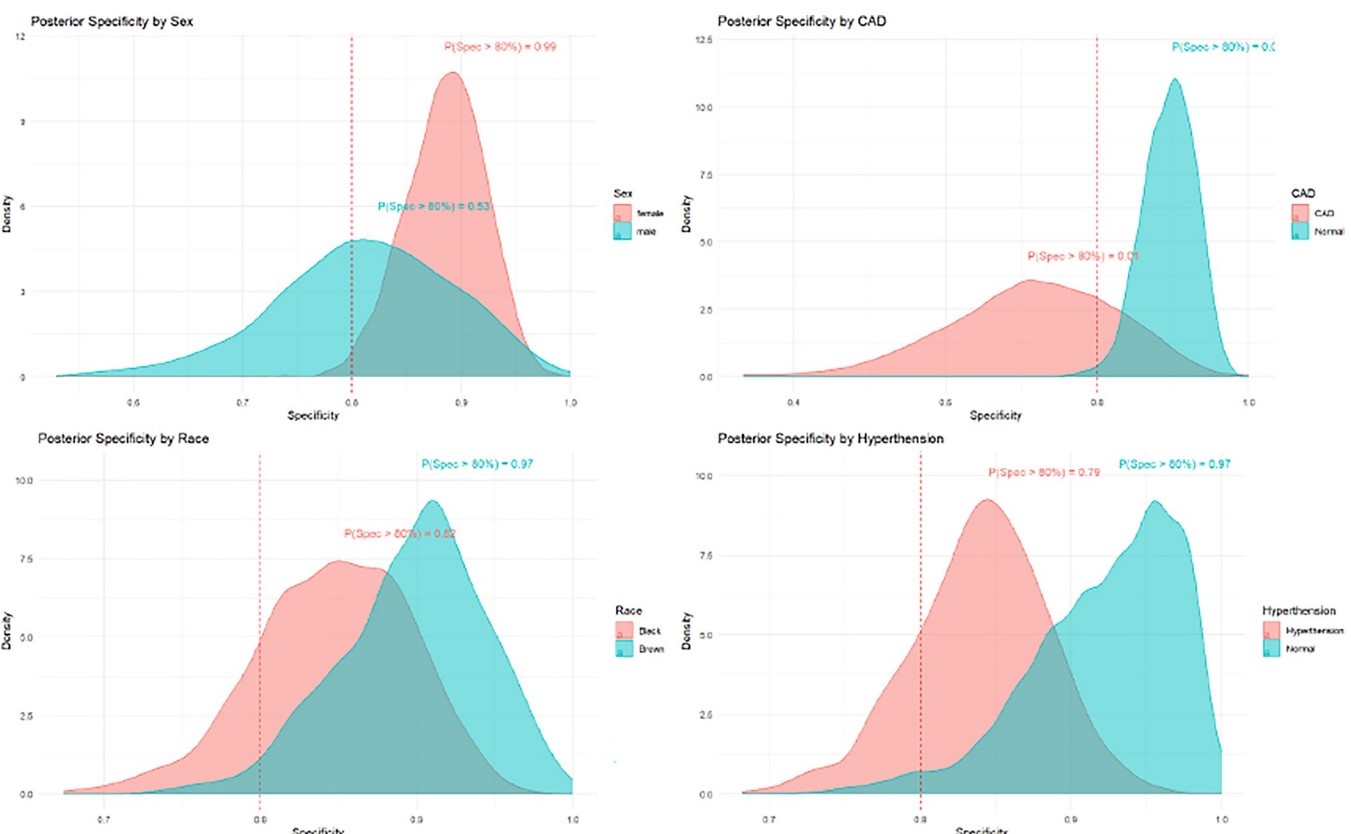

**Fig 3. Posterior probability that the specificity would be higher than 80% in specific subgroups of the population.** From the upper left frame down, sex, CAD, race (self-reported) and hypertension.

diabetes mellitus in a community context. ABIs performance in the considered subgroups of sex, SAH, race and CAD were substantially higher in not-diseased people, females and brown ethnicity (Fig 3).

## Discussion

To the best of our knowledge, this is one of the few studies to evaluate the accuracy of ABI in screening for PAD among DM patients in a community context. We identified a PAD prevalence of 15.98% in a predominantly female population in their sixth decade of life of Black ethnicity, low income, and lacking therapeutic adherence, reflecting the social and demographic characteristics most susceptible as identified in the global literature on PAD. These findings corroborate the relationship between specific social determinants and the disease and highlight the need for special attention to racial, social, and gender disparities in the management of PAD, which have been observed to be significantly underdiagnosed and undertreated [24–26].

The rising global prevalence of DM poses a significant concern and is expected to influence the incidence and prevalence of PAD substantially. Individuals with diabetes are two to four times more likely to develop the disease compared to the general population. Moreover, in this group, PAD presents more severely, progresses more rapidly, tends to affect distal arteries, and is more likely to result in amputation and death [27]. Due to the increased risk of PAD-related complications, non-invasive and accurate vascular assessments of the lower limbs are essential in this population [28].

The guidelines of the International Working Group on Diabetic Foot (IWGDF) emphasize the importance of foot risk assessment, including PAD screening, in individuals with DM at the time of diagnosis, annually after that, upon hospital admission, and if foot problems arise [29]. In asymptomatic individuals, identifying PAD associated with PN or foot deformity categorizes patients as "moderate risk," which should lead to referral to specialized services that will adopt strategies to minimize the risk of foot ulceration and optimize the management of cardiovascular risks [30].

However, PAD is commonly underdiagnosed due to three main factors: (i) a large portion of individuals with PAD are asymptomatic or present atypical symptoms; (ii) there is an underestimation of the disease by healthcare professionals, with studies showing that less than half of the physicians are aware of their patients' PAD diagnoses; (iii) public awareness about PAD is low, with only 26% of adults expressing familiarity with the disease, a lower rate than for other cardiovascular diseases or atherosclerotic risk factors [31, 32].

This situation underscores the importance of correctly interpreting accuracy measures to understand the predictive capacity of diagnostic methods for diseases and determine their appropriate use in various contexts. Sensitivity and specificity alone do not provide predictive measures; they indicate the relationship between the presence of disease and test results. Predictive values depend heavily on the disease's prevalence, limiting their generalization outside the original study unless based on representative random samples [33]. The likelihood ratio (LR), in turn, emerges as a superior metric independent of prevalence, allowing for adjustments in disease probabilities [33]. An LR+ $\geq$ 10 strongly suggests the presence of the disease; a low LR- indicates its absence [34]. Diagnostics are effective when significantly altering the probability of disease, with LR+ > 10 or LR- < 0.1 considered ideal [35].

It is essential to highlight the independence between LR+ and LR-. A test's ability to confirm the disease does not guarantee its effectiveness in excluding it [36, 37]. In our study, using ABI for PAD screening, an LR+ of 14.12 demonstrates that values $\leq$ 0.9 significantly increase the chance of PAD, validating its effectiveness in confirmation. However, an LR- of 0.66, with values between 0.9 and 1.4, only slightly reduces the chance of PAD, indicating lower effectiveness of ABI in excluding the disease. In a population screening context, it becomes less important that the initial test reliably diagnoses PAD, as the consequences of a false-positive result would be less than those of a false-negative result [38]. Minimizing the number of undiagnosed PAD cases is particularly important so that early interventions can be adopted to prevent disease-related severe complications.

In our study, the performance of the ABI was better in populations without CAD, in women, and in brown ethnicity. However, it is known that in patients with significant coronary artery disease and ethnic groups such as African Americans and Hispanics, the presence of PAD can be more pronounced, potentially improving the accuracy of the ABI in its detection. In men, PAD tends to develop at younger ages and with greater severity compared to women, influencing the interpretation of ABI results, as the test accuracy may be better in populations with more evident PAD. Conversely, in populations with diabetic foot ulcers, systemic arterial hypertension, and other risk factors for arterial calcification, the ABI can yield inaccurate results [10, 38].

Optimal medical therapy (OMT) in atherosclerotic disease is crucial to reduce morbidity and mortality associated with this condition. In carotid atherosclerotic disease, reducing delays in evaluation and initiating OMT involving antiplatelets, statins, and antihypertensives in patients with cerebrovascular disease resulted in an 80% decrease in stroke rates [39]. However, patients with PAD receive fewer preventive treatments than those with other cardiovascular conditions, with a notable underuse of statins. Analyzing a broad cohort of individuals

with peripheral disease as their only atherosclerotic disease, 42% of those diagnosed were not using statins, and only 5.8% were on high-intensity statin treatment [40].

The NEtwork to control AtheroThrombosis (NEAT) study revealed that one of the main barriers to prescribing evidence-based therapies in this population was clinical judgment, which, in our analysis, permeates through accurate disease diagnosis [41]. Understanding factors compromising ABI accuracy and implementing methods to enhance disease diagnostic efficacy in the most susceptible group for missed diagnoses are essential to optimize PAD detection in DM patients. The toe-brachial index (TBI) is preferred due to the rare involvement of digital arteries by medial calcification, though its application is limited to research settings. Alternatively, Doppler flow assessment and ankle artery velocities may overcome ABI limitations, enabling occlusive disease identification in the presence of arterial calcification and elevated risk of CVD and limb complications [32].

This study presents several limitations: (i) recruitment conducted in a geographically delimited area with voluntary participant inclusion, potentially limiting result generalization to broader or diverse populations. However, the included population's characteristics align with literature findings, particularly in low- to middle-income countries; (ii) the sample size did not reach the predetermined number, possibly reflecting limited public knowledge about DM and its complications, resulting in less concern and interest in investigation among the target population; (iii) reliance on a single specialist and lack of blinding for index tests and diagnoses may introduce bias risk. However, to minimize potential influences on the results, it was chosen to perform the test and reference measurements on different days. These limitations underscore caution in generalizing this study's results to various clinical contexts and advocate for additional larger-scale studies to identify strategies ensuring more efficient PAD screening and identification of populations at higher cardiovascular and limb adverse outcomes risk.

## Conclusion

In conclusion, this study highlights the complexity of screening for PAD in individuals with DM, underscoring the limitations of the ABI in this context. The importance of careful interpretation of test results is emphasized, as a high LR- can misleadingly suggest the absence of the disease, exposing the at-risk population to underdiagnosis and, consequently, to the undertreatment of PAD. The significant prevalence of the disease and its connection to sociodemographic factors underscore the need for diagnostic and therapeutic strategies that address racial, socioeconomic, and gender disparities. Furthermore, observing discrepancies in prescribing optimal medical therapies in some studies highlights the urgency of overcoming barriers in diagnosis and treatment. Therefore, diagnostic accuracy and the implementation of evidence-based therapies are crucial to mitigate the adverse impact of PAD, especially in vulnerable populations.

## Supporting information

**S1 Dataset.**
(XLSX)

## Acknowledgments

The authors give special thanks to all members of the Vascor Project group for logistical support in conducting the research and to the Residents' Association of Vale do Ogunjá for their collaboration in disseminating the research and recruiting participants.

## Author Contributions

**Conceptualization:** Monique Magnavita Borba da Fonseca Cerqueira.

**Data curation:** Monique Magnavita Borba da Fonseca Cerqueira, Dandara Almeida Reis da Silva, Dario Gregori, Lucélia Batista Neves Cunha Magalhães, Magno Merces Weyll Pimentel.

**Formal analysis:** Monique Magnavita Borba da Fonseca Cerqueira, Dandara Almeida Reis da Silva, Lucélia Batista Neves Cunha Magalhães, Magno Merces Weyll Pimentel.

**Investigation:** Neiva Sueli Santana Gonçalves Bastos, Dandara Almeida Reis da Silva, Lucélia Batista Neves Cunha Magalhães, Magno Merces Weyll Pimentel.

**Methodology:** Monique Magnavita Borba da Fonseca Cerqueira, Dario Gregori, Magno Merces Weyll Pimentel.

**Project administration:** Monique Magnavita Borba da Fonseca Cerqueira.

**Software:** Dario Gregori.

**Supervision:** Monique Magnavita Borba da Fonseca Cerqueira, Magno Merces Weyll Pimentel.

**Validation:** Magno Merces Weyll Pimentel.

**Visualization:** Magno Merces Weyll Pimentel.

**Writing – original draft:** Monique Magnavita Borba da Fonseca Cerqueira, Dandara Almeida Reis da Silva, Magno Merces Weyll Pimentel.

**Writing – review & editing:** Monique Magnavita Borba da Fonseca Cerqueira, Neiva Sueli Santana Gonçalves Bastos, Dandara Almeida Reis da Silva, Dario Gregori, Lucélia Batista Neves Cunha Magalhães, Magno Merces Weyll Pimentel.

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
