## [Decision Letter · Decision Letter 0]

26 Jun 2024

PONE-D-24-20537Accuracy of Ankle-Brachial Index in Screening for Peripheral Arterial Disease in People with DiabetesPLOS ONE

Dear Dr. Cerqueira,

Thank you for submitting your manuscript to PLOS ONE. After careful consideration, we feel that it has merit but does not fully meet PLOS ONE’s publication criteria as it currently stands. Therefore, we invite you to submit a revised version of the manuscript that addresses the points raised during the review process.

We look forward to receiving your revised manuscript.

Kind regards,

Biswabandhu Jana, Phd

Academic Editor

PLOS ONE

Journal Requirements:

Additional Editor Comments:

In 217, this is the first study to evaluate the accuracy of ABI screening for PAD among DM patients. Please check other studies may have done that kind of researches. You may see the link 10.4103/jfmpc.jfmpc_1546_20.

Novelty of this work is not clear.

What is taken as ground truth data.

How different types of blood flow wave affects in the measurements.

Reviewers' comments:

Reviewer's Responses to Questions

**Comments to the Author**

1. Is the manuscript technically sound, and do the data support the conclusions?

Reviewer #1: Yes

Reviewer #2: Partly

2. Has the statistical analysis been performed appropriately and rigorously? 

Reviewer #1: Yes

Reviewer #2: No

3. Have the authors made all data underlying the findings in their manuscript fully available?

Reviewer #1: Yes

Reviewer #2: Yes

4. Is the manuscript presented in an intelligible fashion and written in standard English?

Reviewer #1: Yes

Reviewer #2: Yes

5. Review Comments to the Author

Reviewer #1: Good contribution to existing knowledge.

Take a cursory look at the Methodology and result section.

Line 50: There should be a reference after elsewhere

Line 68: Do you expect a screening tool to detect all cases, (modify)

What specifically differentiates your cohort of subjects from the community from those done in hospital settings (Elaborate more on this to justify the study)

Line 92/93: Refusal to participate is not an exclusion criteria ditto for the inclusion criteria of agreeing to participate (Expunge)

Table 1 : Check the number and percentage of brown/black and correct as appropriate

Also check the common condition percentage and number

Can you present information on true positives, true negatives, false positive and false negative before Table 2

What is the mean/median ABI from this study

Reviewer #2: The article is well informative, however authors should consult some article related to their research from a high impact factor journal and should present results, methods and discussion in detail accordingly as a sample, especially include more statistics and inclusion of figures for data presentation, otherwise include more parameters, otherwise article looks below standard of the journal.

6. PLOS authors have the option to publish the peer review history of their article (what does this mean?). If published, this will include your full peer review and any attached files.

Reviewer #1: No

Reviewer #2: **Yes: **Sarfraz Ahmed

---

## [Author Response · Author response to Decision Letter 0]

31 Jul 2024

Dear Dr. Biswabandhu Jana,

We sincerely appreciate your meticulous review of our manuscript. We would like to clarify a few points in response to your comments.

Initially, we corrected the statement that our study would be the first with this focus after thoroughly reviewing the works included in the systematic reviews available. In the latest systematic review published by Chuter et al. (2024), we observed that a recent study did not clarify whether the participants were recruited from the community, while another could not be thoroughly reviewed. Our research group did not conduct a systematic review on the subject, so we modified this sentence. The study referenced in link 10.4103/jfmpc.jfmpc_1546_20, which has similar characteristics to ours, is a cross-sectional observational study conducted during a workshop for diabetic patients in the outpatient department of a hospital, without specifying whether the participants were already being followed or were recruited from the community.

We believe that our study contributes to the existing literature by investigating the performance of the ABI in screening PAD in a community context, where the characteristics of recruited participants differ considerably from specialized centers, even among asymptomatic individuals. In study design, the recruitment location of participants is crucial, as an overly specific selection may not adequately represent the general population, compromising the applicability of the results in screening tests. It is known that in specialized centers, the prevalence of the disease may differ substantially from that found in the community population. More severely affected patients may be included in areas with higher disease prevalence, which may overestimate the test's performance.

Additionally, based on the systematic reviews published on the topic, we identified that the current evidence on the role of ABI <0.9 as a potentially helpful test for diagnosing PAD is based on a limited number of studies (n = 3), with moderate to high risks of bias. Therefore, we believe there is a demand for studies with better-defined methodologies to expand our understanding of the performance of this test, considering the significant impact of undiagnosed and inadequately managed PAD in individuals with diabetes.

Sincerely,

The Authors.

Dear Reviewer #1, 

We appreciate your careful review of our manuscript. We have reviewed all the texts and accepted and incorporated your correction suggestions. In the revised manuscript, these changes are appropriately highlighted.

Regarding your question about how our study population differs from populations in studies conducted in hospital settings, we would like to clarify a few points. The accuracy of a diagnostic tool depends not only on its sensitivity and specificity but also on the prevalence of the disease in the studied population. In studies conducted in specialized centers, the prevalence and severity of the disease significantly differ from the community population, which can lead to an overestimation of the test's performance.

This justification has been better elaborated and is available in the introduction of the revised work.

Sincerely,

The authors.

Dear Reviewer #2, 

We appreciate your careful review of our manuscript. We have reviewed the entire article. To enhance the presentation of our results, in addition to evaluating the diagnostic accuracy of the text, we have added a Bayesian regression model to explore the diagnostic performance of ABIs in specific population subgroups. The detailed analyses are described in the Methods section, and their graphical representations have been included as new figures in the article.

Sincerely,

The authors.

---

## [Editor Report · Decision Letter 1]

6 Aug 2024

Accuracy of Ankle-Brachial Index in Screening for Peripheral Arterial Disease in People with Diabetes

PONE-D-24-20537R1

Dear Dr. Cerqueira,

We’re pleased to inform you that your manuscript has been judged scientifically suitable for publication and will be formally accepted for publication once it meets all outstanding technical requirements.

Kind regards,

Biswabandhu Jana, PhD

Academic Editor

PLOS ONE
---

## [Editor Report · Acceptance letter]

10 Oct 2024

PONE-D-24-20537R1 

PLOS ONE

Dear Dr. Cerqueira, 

I'm pleased to inform you that your manuscript has been deemed suitable for publication in PLOS ONE. Congratulations! Your manuscript is now being handed over to our production team.

Kind regards, 

on behalf of

Dr. Biswabandhu Jana 

Academic Editor

PLOS ONE